# Flipping Against All Odds: Reducing LLM Coin Flip Bias via Verbalized Rejection Sampling

## Abstract

Large language models (LLMs) can often accurately describe probability distributions using natural language, yet they still struggle to generate faithful samples from them. This mismatch limits their use in tasks requiring reliable stochasticity, such as Monte Carlo methods, agent-based simulations, and randomized decision-making. We investigate this gap between knowledge and sampling in the context of Bernoulli distributions. We introduce Verbalized Rejection Sampling (VRS), a natural-language adaptation of classical rejection sampling that prompts the LLM to reason about and accept or reject proposed samples. Despite relying on the same Bernoulli mechanism internally, VRS substantially reduces sampling bias across models. We provide theoretical analysis showing that, under mild assumptions, VRS improves over direct sampling, with gains attributable to both the algorithm and prompt design. More broadly, our results show how classical probabilistic tools can be verbalized and embedded into LLM workflows to improve reliability, without requiring access to model internals or heavy prompt engineering.

## 1 Introduction

Large language models (LLMs) have demonstrated remarkable capabilities in generating coherent text and even performing reasoning tasks. An emerging question is whether LLMs can understand and reproduce probabilistic processes when prompted in natural language. In particular, if we ask an LLM to behave like a random sampler for a known distribution (e.g., produce coin flip outcomes with a given probability), will it faithfully do so? Reliable sampling underpins Monte Carlo algorithms [13, 19], probabilistic programming [4], agent-based simulations [11, 3], and randomized decision making [16, 15]; yet, despite randomness being central to modern computation, the extent to which contemporary LLMs can generate faithful i.i.d. samples remains largely unexplored.

Recent work has begun to study LLMs not just as next-word predictors but as generators of random outcomes drawn from specified distributions. Empirical evidence shows that, while LLMs can infer probability distributions [6] and do Bayesian updates to approximately infer a coin's bias when given data [7], their own samples from a distribution remain biased [11]. Figure 1(a;b) illustrate this gap for Bernoulli distributions. Hence, LLMs know what a fair coin is, but they struggle to behave like one.

This mismatch poses concrete risks from a user's perspective. A user who sees an LLM accurately reasoning about a distribution might trust it to sample from that distribution; hidden bias can then contaminate downstream workflows, skew survey simulators, or introduce unfairness in stochastic tie-breakers. If an LLM cannot flip a fair coin, could it be trusted to sample from more complex distributions? This raises safety, reliability, and fairness concerns across the stack.

Submitted to 39th Conference on Neural Information Processing Systems (NeurIPS 2025). Do not distribute.

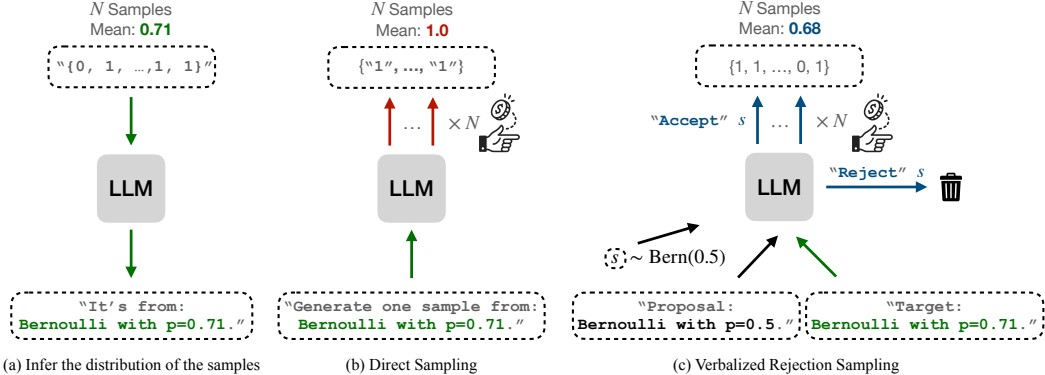

Figure 1: Illustrations of the knowledge-sampling gap and two different sampling methods.

In the setting of Bernoulli distributions, we present a comprehensive study of correcting LLM sampling bias via a language-adapted rejection-sampling framework, and uncover surprising interactions between prompt design and algorithmic guarantees. Our contributions include:

- **Sampling Faithfulness Study (Section 4).** We measure how faithfully LLMs generate i.i.d. Bernoulli samples when prompted directly. Across four models, sampling bias varies significantly with the phrasing of the distribution. Chain-of-thought only helps in some cases. We also quantify the gap between a model's ability to identify a distribution and its ability to simulate it.

- **Verbalized Rejection Sampling (VRS) (Section 5).** We adapt the classical rejection sampling method through natural language into LLMs. VRS is model-agnostic (for both open-source and proprietary LLMs), requires no access to the model weights, and keeps the LLM in a black-box. Given a fixed prompt with textual descriptions of the target and proposal distributions alongside a candidate sample, the LLM is instructed to perform the accept/reject step. Our empirical study shows a significant reduction of the bias for the samples.

- **Empirical and Theoretical Insights (Section 6).** Effectively, VRS draws a Bernoulli random variable to decide whether to accept a proposed sample. Counter-intuitively, this indirection produces less sampling bias than prompting the model to output a sample directly. We analyze this phenomenon theoretically, proving—under mild assumptions—that VRS can generate samples with less bias than direct sampling and separating the gains attributable to the prompt phrasing from those guaranteed by the algorithm itself.

Beyond correcting the specific failure mode of Bernoulli sampling, our study opens a broader path towards integrating principled randomness into LLM-based systems. Faithful Bernoulli generation is a basic requirement for reliable LLM-driven simulations and stochastic reasoning. Our results show that a lightweight, theoretically sound wrapper—without model access or hyper-parameter tuning—substantially narrows the knowledge-sampling gap. More broadly, our work illustrates how classical statistical tools can be verbalized and paired with LLMs to deliver reliability without resorting to opaque prompt engineering.

## 2 Related Work

**Sampling and flipping coins with LLMs.** Recent work shows that LLMs often exhibit a gap between knowing and sampling from a distribution. [6] find that LLMs can describe the target probabilities, yet when asked to "roll a die" or "flip a coin" their outputs exhibit large bias. They show that incorporating code generation with Python tool use can alleviate the problem. In contrast, we focus on improving sampling within the natural language space, leveraging LLMs' inherent probabilistic reasoning capabilities. While one could bypass the model to obtain true samples from a target distribution, enabling LLMs to faithfully perform such tasks themselves is both practically useful and scientifically insightful. [14] explore how LLMs "flip a fair coin" and "flip 20 fair coins". They find that current LLMs not only replicate human biases but often amplify them. [7] probe the online learning setting of Bernoulli distribution from a Bayesian inference angle. They show that with sufficient in-context examples, LLMs update their estimate of a coin's bias roughly following Bayes' rule. Unlike their focus on online learning and belief updating, we do not assume sequential

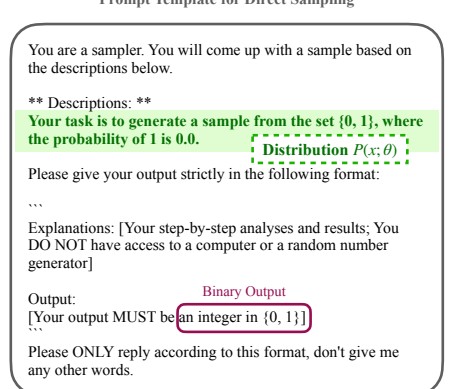
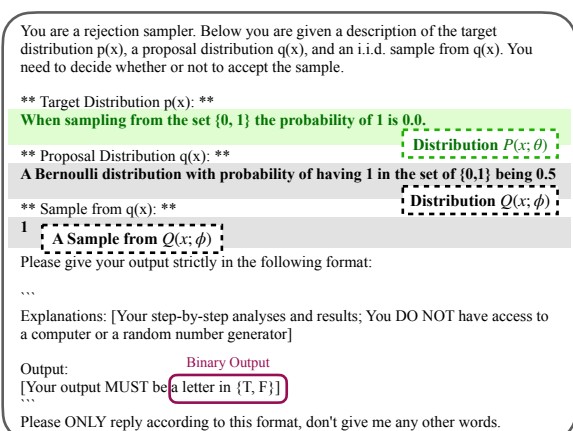

Figure 2: Prompt templates for direct sampling and Verbalized Rejection Sampling.

access to data and instead concentrate on the generation of i.i.d. samples from a fixed Bernoulli distribution. [11] and [8] find similar gaps in settings beyond Bernoulli (e.g., poll simulation, categorical distribution), showing that LLMs can summarize distributions but fail to sample from them reliably, echoing the Bernoulli findings on a higher-dimensional setup. Together, these studies reveal a recurring pattern: LLMs know the right distributions but struggle to sample from them faithfully. Our work aims to reduce this mismatch by adapting the rejection sampling algorithm to LLMs, leveraging their internal probabilistic behavior to guide natural language based sampling.

**Natural language and text based parameterization.** Recent work explores using natural language to parameterize models, treating LLMs as inference engines that interpret and evaluate these descriptions. This makes model specification more accessible and interpretable. [12] introduce LLM Processes, where LLMs generate predictive distributions conditioned on natural language inputs and in-context data. Their method operates in an in-context, non-parametric style and requires access to token logits. In contrast, we treat language as a parametric description of a fixed distribution, without past data or logit access. [18] propose Verbalized Machine Learning (VML), where prompts act as natural language parameters for deterministic functions. Our work instead focuses on probabilistic distributions and faithful sampling. [2] presents a theoretical framework demonstrating that a finite set of function compositions, analogous to a vocabulary, can approximate any continuous mapping, drawing parallels between linguistic compositionality and function approximation. These studies underscore the potential of natural language as a medium for specifying probabilistic models. In our work, we focus on the Bernoulli distribution as a fundamental case study, demonstrating how LLMs can be guided to generate faithful samples from a simple yet foundational probabilistic model.

# 3 Problem Setup

Our investigation focuses on the ability of LLMs to generate faithful i.i.d. samples from distributions described purely in natural language. Focusing on Bernoulli distributions, defined by a single numerical parameter $p \in [0, 1]$, we treat LLMs as samplers accessed solely through text interaction.

## 3.1 Parameterizing Distributions in Natural Language

In our setting, the distribution is parameterized by a textual prompt. Formally, we denote this natural language parameterized distribution as $P(x; \theta)$, where $\theta$ captures both the underlying numerical parameter $p$ and the linguistic phrasing of the prompt. Figure 2(left) shows an example, where

$P(x; \theta) =$ *"Your task is to generate a sample from the set {0, 1}, where the probability of 1 is 0.0."*.

For the same $p$, different phrasings may lead to different sampling behaviors. We test several ways of phrasing a Bernoulli distribution, and write $P(x; p)$ for a fix phrasing. For each phrasing, we test 101 values of $p \in \{0.0, 0.01, 0.02, \ldots, 1.0\}$. For each $p$, we query the LLM 100 times independently with the same prompt, and extract the binary output (i.e., '0' or '1') to form the resulting i.i.d. samples.

 ### 3.2 LLMs as Black-Box Samplers

We treat LLMs as black-box samplers, accessed solely via APIs. The only controllable input is the prompt; the only observable output is text. For open-source models, we use vLLM [9], but we assume no access to internals such as weights, activations, or token-level logits. This contrasts with prior work [7, 8, 12] that uses output token logits to estimate sampling probabilities.

This API-only setup allows consistent evaluation across both open-source and proprietary models, reflecting realistic usage where internals are inaccessible. It also better supports techniques like chain-of-thought (CoT; [17]) prompting, which can distort token-level probabilities by conditioning on generated reasoning: with CoT, logits reflect $p(x \mid \text{reasoning for } x)$ instead of the intended $p(x)$. We also fix all decoding hyperparameters (e.g., temperature, top-k) to their default values given in the API, since most real world users do not adjust them, and often do not have the ability to do so.

## 4  How Reliable is Direct Sampling?

This section examines the reliability of direct sampling from LLMs. We first compare their ability to generate samples to their ability to recognize distributions, then explore how prompt phrasing affects sampling bias, and finally test whether chain-of-thought reasoning improves sample quality.

### 4.1  Measuring the Knowledge-Sampling Gap

To assess the gap between an LLM's understanding of a Bernoulli distribution and its ability to sample from it, we compare its evaluative and generative performance in a controlled setup, using Llama-3.1-70B-Instruct [5]. We first test the model's ability to identify the correct Bernoulli distribution from data. For 11 equally spaced probabilities $p_0, ..., p_{10}$, s.t. $p_i \in [0, 1]$, we generate 100 i.i.d. samples using Python, forming datasets $S_i$. For each pair $(i, j)$, we prompt the LLM to decide whether $S_i$ was drawn from $\text{Bern}(p_j)$, producing an $11 \times 11$ response matrix. Diagonal entries should be "Yes", off-diagonals "No". We repeat this process five times and report average accuracies in Figure 3(a). We then test the model's sampling behavior by prompting it to generate 100 samples for each $p_i$, using the template in Figure 2(left). The resulting sets $\hat{S}_i$ are evaluated using the same method as before. The average accuracies over five runs are reported in in Figure 3(b).

The left panel shows high off-diagonal accuracy for Python generated data (i.e., confidently rejecting incorrect hypotheses), with minor errors along the diagonal due to natural sample variation (e.g., 48 ones out of 100 for $p = 0.5$ may lead to confusion with $p = 0.48$, hence, rejecting the correct hypotheses). In contrast, the right panel shows major degradation for LLM-generated samples. Diagonal accuracy drops significantly for all $p_i$, except the edge cases when $p = 0.0$ and $p = 1.0$. Moreover, we observe an asymmetry in the off-diagonal entries: the lower triangle of the matrix exhibits much worse accuracy than the upper triangle. This indicates that samples from $p_i$ are often misclassified as having come from $p_j$ with $j > i$, suggesting that the LLM-generated samples are consistently biased toward ones. These results reveal a clear knowledge–sampling gap: LLMs can evaluate distributions well but fail to sample from them faithfully. Unlike question answering, where each input has a correct target, i.i.d. sampling lacks per-instance ground truth, making it a fundamentally different and underexplored capability.

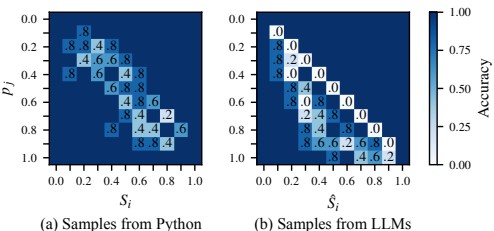

(a) Samples from Python  (b) Samples from LLMs

Figure 3: Recognition accuracy matrix.

### 4.2  How Much Can Prompt Phrasing Reduce Sampling Bias?

The previous section used a single fixed phrasing to describe the Bernoulli distribution (see Figure 2, left). Yet, natural language allows many equivalent ways to express the same distribution, raising the question: how much can phrasing affect sampling bias? In the prior setup, the prompt emphasized the probability of generating a 1, denoted $P_1(x; p)$, as illustrated in Figure 4(b). Notably, this formulation focuses solely on the probability of generating a 1, which may partly explain the tendency of the model to produce more 1s than 0s in the in the sampled outputs.

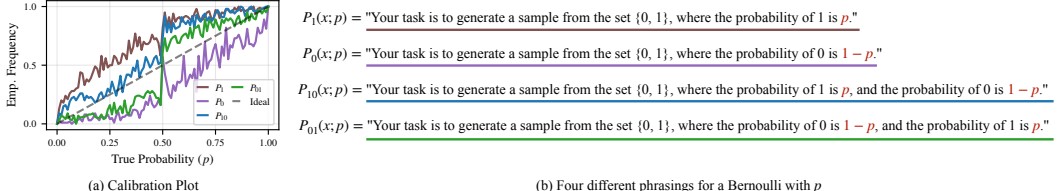

(a) Calibration Plot            (b) Four different phrasings for a Bernoulli with $p$

Figure 4: Calibration plots for direct sampling and the four different phrasings.

To explore this, we test three alternative phrasings that shift or balance the focus across outcomes, as shown in Figure 4(b). For each, we sample across a range of $p$ values using Llama-3.1 and plot the empirical frequency of 1s against the ground truth, yielding calibration curves shown in Figure 4(a). The calibration curves show that the balanced descriptions, i.e., those stating both probabilities, yield samples that are better calibrated. Nevertheless, all four phrasings result in noticeable bias. This result resonates with [1], where they found that when prompting humans to imagine a coin flip, mentioning only 'heads' or mentioning only 'tails' will lead to a similar sampling bias.

**Quantitative comparison using Sum of TV Distance (STVD).** To quantify the calibration performance of different phrasings, we compute the area between each calibration curve and the ideal diagonal reference line. Specifically, for each $p_i$, we calculate the absolute difference between the empirical sampling frequency $\tilde{p}_i$ and the true value $p_i$, and sum these over all 101 values, i.e., $\text{STVD} = \sum_{i=0}^{100} |\tilde{p}_i - p_i|$. Since this absolute difference corresponds to the total variation (TV) distance between two Bernoulli distributions, we refer to the resulting metric as the Sum of TV Distances (STVD) where smaller is better. See Appendix A.1 for more details about the TV distance.

Table 1 presents the STVD values for the four phrasings under direct sampling. For Llama 3.1, the best-performing phrasing $P_{01}$ achieves an STVD of 11.08, nearly half that of the baseline $P_1$, which scores 21.80. We also include results for other LLMs, including GPT-4.1-nano, DeepSeekV3 [10], and Qwen-2.5 72B [20]. Interestingly, the best-performing phrasing varies across models, as highlighted by the underlined entries. The calibration plots for the other models can be found in Appendix B.1.

These findings suggest that while prompt design can influence sampling bias, relying solely on prompt engineering to eliminate bias can be difficult and inconsistent across model family, and additional mechanisms are likely needed for more systematic approaches to correct sampling bias.

Table 1: Quantitative comparison between Direct Sampling and VRS in STVD ($\downarrow$).

| Method | Llama-3.1 70B | | | | | GPT-4.1-nano | | | | | DeepSeekV3 | | | | | Qwen-2.5 72B | | | | |
|---|---|---|---|---|---|---|---|---|---|---|---|---|---|---|---|---|---|---|---|---|
| | $P_1$ | $P_0$ | $P_{10}$ | $P_{01}$ | mean | $P_1$ | $P_0$ | $P_{10}$ | $P_{01}$ | mean | $P_1$ | $P_0$ | $P_{10}$ | $P_{01}$ | mean | $P_1$ | $P_0$ | $P_{10}$ | $P_{01}$ | mean |
| Direct | 21.80 | 17.95 | 12.10 | 11.08 | 15.73 | 17.87 | 30.23 | 16.63 | 19.24 | 21.00 | 17.76 | 19.39 | 20.78 | 23.26 | 20.30 | 20.73 | 18.72 | 19.00 | 22.64 | 20.27 |
| VRS | 5.91 | 7.63 | 5.52 | 5.56 | **6.20** | 12.96 | 13.06 | 9.50 | 8.46 | **11.00** | 5.34 | 9.06 | 5.29 | 6.94 | **6.66** | 5.93 | 6.35 | 4.49 | 5.12 | **5.47** |

## 4.3 Does Chain-of-Thought (CoT) Help Sampling?

Since phrasing alone does not eliminate sampling bias, we explore whether modifying the instruction for the output can help. Prior work [14, 7, 8, 12] often asks LLMs to output the sample immediately, enabling access to token logits for estimating predictive distributions. However, this approach is constrained to open-source models and treats LLMs more as likelihood models than samplers. In our setting, we only use LLMs for sampling and do not require access to logits or early output. This allows us to apply CoT [17] prompting, where the model first generates reasoning before giving its final answer. While sampling differs from question answering, CoT may increase output variability by encouraging diverse reasoning paths, potentially reducing bias.

To test this, we instruct the model to produce reasoning of varying lengths $N$ (ranging from 0 to 500 words) before answering, along to an 'Auto' setting where no length constraint is imposed. The 'Auto' is the default setting for experiments in previous sections, which uses the template in Figure 2(left). For different $N$, we modify the '*Explanations*' instruction in the prompt template to include a sentence saying that '*Your analysis must have around N words*'.

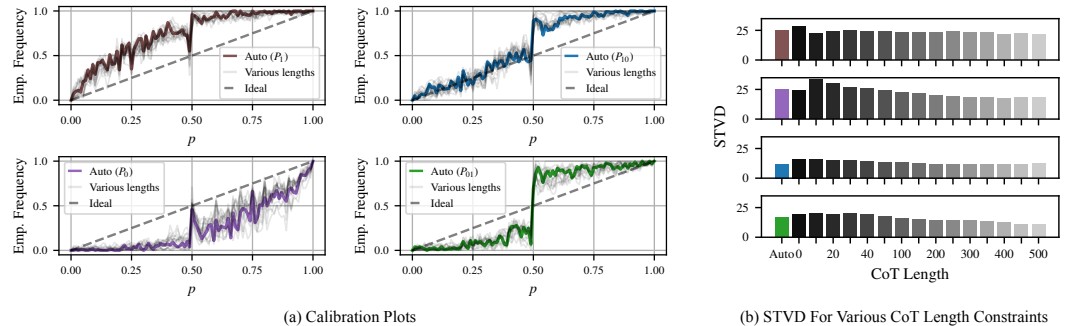

(a) Calibration Plots

(b) STVD For Various CoT Length Constraints

Figure 5: Calibration plots and STVD trend for various reasoning length constraints.

Figure 5 presents the calibration plots (left) and STVD scores (right) for Llama-3.1 under different CoT length constraints. Overall, reasoning length has limited effect on bias, though longer CoT slightly improves calibration. Direct output without reasoning often performs worse than the 'Auto' setting. However, this pattern does not hold across models. As shown in Figure 6, GPT-4.1 and Qwen2.5 show no consistent improve-

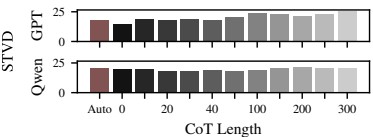

Figure 6: STVD vs CoT Length.

ment with longer CoT; in some cases, STVD increases as reasoning length grows. These mixed results suggest that, unlike in question answering, CoT is not a reliable method for reducing sampling bias, and its effect is model-dependent. For consistency, we use 'Auto' in all remaining experiments.

## 5 Verbalized Rejection Sampling

In the previous section, we explored ways to reduce sampling bias through prompt phrasing and instruction design. While these strategies do influence the behavior of LLMs, the results suggest that prompt-only interventions are insufficient for reliably eliminating bias. If direct sampling cannot be fully corrected through language alone, we may instead embrace the bias and mitigate it using algorithmic techniques. In probabilistic methods, several algorithms exist to transform biased proposals into unbiased samples. One such method is rejection sampling, which generates candidate samples from a proposal distribution and selectively accepts them to match a desired target distribution. In the remainder of this section, we adapt rejection sampling to operate entirely within the language interface of LLMs, and we refer to this method as verbalized rejection sampling (VRS).

### 5.1 Rejection Sampling

Rejection sampling is a sampling technique to generate samples from a target distribution $P$ while only having access to samples from a (typically simpler) proposal distribution $Q$. We assume that both $P$ and $Q$ can be evaluated (but only $Q$ can be directly sampled from). The general idea is that we can generate a sample from $P$ by instead sampling from $Q$ and accepting the sample with probability $P(x)/(MQ(x))$ where $M < \infty$ is a bound on the ratio $P(x)/Q(x)$. We assume that both $P$ and $Q$ are Bernoulli distributions with parameters $p$ and $q$. In this case, we can compute $M$ analytically as: $M = \max\{p/q, (1-p)/(1-q)\}$. Let $A(x)$ denote the acceptance probability of $x \sim Q$ which is

$$A(x) = \begin{cases} \frac{P(x)}{MQ(x)} = \frac{p}{Mq} & \text{if } x = 1 \\ \frac{P(x)}{MQ(x)} = \frac{1-p}{M(1-q)} & \text{if } x = 0 \end{cases}. \tag{1}$$

The accept/reject step effectively draws a sample from $\text{Bern}(A(x))$. The overall acceptance rate is $\alpha = \sum_{x \in \{0,1\}} Q(x)A(x) = 1/M$. See Appendix A.2 for more details about rejection sampling.

### 5.2 Adapting Rejection Sampling to LLMs

Figure 1(c) illustrates the overall idea behind VRS. Classical rejection sampling requires three inputs: the target distribution $P$, the proposal distribution $Q$, and a sample $x \sim Q$. The algorithm evaluates

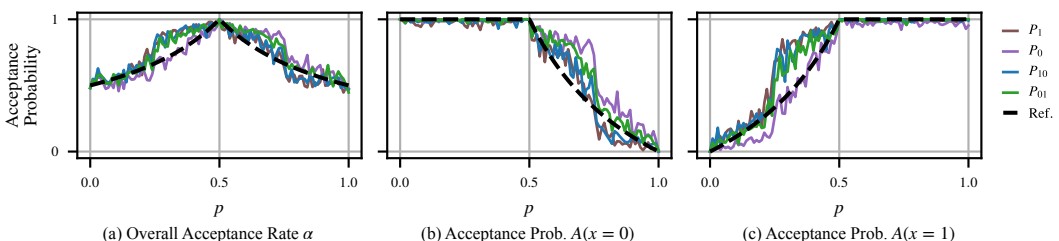

Figure 8: Empirical acceptance rates for VRS.

whether to accept or reject $x$ based on these inputs, returning a binary decision. To implement this in the LLM setting, we design a prompt template (Figure 2, right) that verbalizes all three components, i.e., descriptions of $P, Q$, and the proposed sample $x$, as natural language. These are inserted into fixed slots in the template. The model is instructed to reason through its decision and then output a single letter from $\{T, F\}$, indicating whether to accept (T) or reject (F) the sample. We send the completed prompt to the LLM and parse its response. If the response indicates acceptance, we retain the sample; otherwise, we generate a new proposed sample and repeat the process. This loop continues until we collect the required number of accepted samples.

## 5.3 Experiments

We evaluate VRS on four different LLMs: Llama-3.1, GPT-4.1-nano, DeepSeekV3, and Qwen-2.5. For each model, we run VRS until it accepts 100 samples for each of the 101 values of $p \in [0.0, 1.0]$, following the same setup as in the direct sampling experiments. As the proposal distribution $Q$, we fix it to a uniform Bernoulli with $q = 0.5$ across all values of $p$. The resulting calibration plot for Llama-3.1 is shown in Figure 7, and the corresponding STVD scores across all models are included in Table 1. The calibration plots for other three LLMs can be found in Appendix B.2.

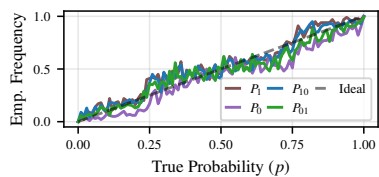

Figure 7: Calibration plot for VRS

Comparing the calibration plot for VRS (Figure 7) with that of direct sampling (Figure 4a), we observe a significant reduction in sampling bias. Across all four prompt phrasings, the calibration curves under VRS closely align with the ideal diagonal reference, indicating much improved fidelity to the target Bernoulli distributions. Figure 8 shows the corresponding empirical acceptance probabilities, which seem to align well with the analytical targets. The improvement is also reflected quantitatively in Table 1: the STVD scores for VRS are substantially lower than those for direct sampling, with most cases showing a reduction of over 50%. In some instances, STVD drops to nearly 25% of the original value. Crucially, this improvement holds across all four LLMs tested (i.e., Llama-3.1, GPT-4.1-nano, DeepSeekV3, and Qwen-2.5), demonstrating that VRS consistently mitigates bias and does so independently of the underlying model.

# 6 Why Does Verbalized Rejection Sampling Work?

The effectiveness of VRS in reducing sampling bias is surprising at first glance since, internally, VRS still relies on the LLM to perform a Bernoulli trial, i.e., deciding whether to accept or reject a sample, which is precisely the type of stochastic behavior we have shown LLMs to struggle with.

*If LLMs are biased in direct sampling, why does wrapping the decision in rejection sampling help?*

Is the improved calibration a result of the specific prompt design used in VRS? Or does the rejection sampling algorithm itself introduce structural guarantees that correct bias, even when implemented via a biased LLM? The remainder of this section explores these possibilities empirically and theoretically.

## 6.1 Is It the Magic in the Prompt?

To investigate whether VRS's improvement stems purely from prompt design, we remove external randomness by fixing the proposed sample to a constant, i.e., $x = 1$. In this case, a faithful LLM

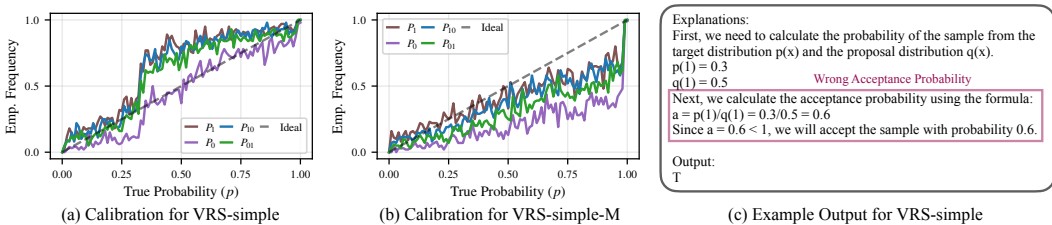

Figure 9: Calibration plots for two ablations and an example LLMs output for VRS-simple.

should accept with probability $A(1)$, as defined in Equation (1). We compare this with the empirical
acceptance probability $\tilde{A}(1)$, estimated from the LLM's responses. Figure 8(c) shows $\tilde{A}(1)$ for
various $p$, using a fixed proposal $Q = \text{Bern}(0.5)$. For the trivial case $p > 0.5$, the alignment is strong.
For $p < 0.5$, the results appear reasonable overall but show a consistent bias, particularly in the range
$p \in [0.2, 0.5]$. To compare more directly with direct sampling, we evaluate $\tilde{A}(1)$ over 101 equally
spaced values of $A(1)$, using the inverse of Equation (1) to recover the corresponding $p$. For each, we
generate a VRS prompt with the computed $p$, a fixed $Q = \text{Bern}(0.5)$, and a fixed sample $x = 1$. We
refer to this setup with fixed proposal and no introduced randomness as *VRS-simple*. If prompt design
alone explains the improvement, VRS-simple should outperform direct sampling in calibration.

Figure 9(a) shows the calibration plot for VRS-simple using
Llama-3.1. Compared to direct sampling (Figure 4a), the re-
sults are slightly more calibrated. Table 2 confirms this, with
the mean STVD dropping from 15.73 to 11.85. This suggests
the VRS prompt helps reduce bias for direct sampling. How-
ever, VRS-simple relies on explicitly computing the inverse
of Equation (1) to tailor the prompt to each target $p$, and the
improvement remains modest compared to full VRS.

Table 2: Ablation STVD ($\downarrow$)

| Method | $P_1$ | $P_0$ | $P_{10}$ | $P_{01}$ | mean |
|---|---|---|---|---|---|
| Direct | 21.80 | 17.95 | 12.10 | 11.08 | 15.73 |
| VRS | 5.91 | 7.63 | 5.52 | 5.56 | 6.20 |
| VRS-simple | 15.83 | 6.55 | 13.97 | 11.05 | 11.85 |
| VRS-simple-M | 11.43 | 29.08 | 13.45 | 19.52 | 18.37 |
| VRS-M | 4.97 | 11.74 | 5.91 | 7.03 | 7.41 |

**Magic or Mirage?** To further understand why the VRS prompt improves sampling, we examine
whether its structure encourages the model to reason differently. One hypothesis is that phrasing the
sampling task in the context of rejection sampling prompts the LLM to internally compute acceptance
probabilities, potentially disrupting its default biases learned during pretraining. To test this, we
manually analyzed the model's reasoning outputs from VRS-simple (see Figure 9(c)). We found
that, while the model often tries to derive the acceptance probability, it frequently does so incorrectly.
In the non-trivial cases where $A(x) \neq 1$, the model tends to compute only the ratio $P(x)/Q(x)$,
omitting the constant $M$ in the denominator.

*Could this incorrect derivation be the reason behind the improvement?* To test that, we designed
variants of VRS-simple and VRS where we explicitly instruct the model to compute and use $M$
correctly. We refer to these as VRS-simple-M and VRS-M, respectively. The calibration plot for
VRS-simple-M is shown in Figure 9(b), with corresponding STVD scores in Table 2. Through
output inspection, we verified that the LLM now correctly computes the constant $M$ in its reasoning.
However, this correction leads to worse performance: the mean STVD increases to 18.37, higher
than in direct sampling. For the full VRS setup, adding the $M$-instruction also results in a slight
degradation, with STVD rising from 6.20 to 7.41, though still outperforming direct sampling.

These results suggest that the improvement from the VRS prompt is not due to accurate computation
of the acceptance probability. Instead, the prompt seems to help in an unexpected way, but it
alone cannot explain the full benefit. The remaining gains likely come from the rejection sampling
mechanism itself, rather than prompt phrasing alone.

## 6.2 Is the Improvement from the Algorithm?

Prompt design alone cannot fully explain the gains from VRS. To analyze the role of the algorithm
itself, we model the LLM as a biased Bernoulli sampler. In VRS, this means the acceptance decision
is not sampled from the true probability $A(x)$, but from a perturbed version $\tilde{A}(x) = A(x) + e(x)$,
where $e(x)$ represents the model's bias. Based on this, we can derive the following proposition.

**Proposition 1** (Informal, see Proposition 1 in Appendix A.3.)**.** *Let $P(x; p), Q(x; q)$ be Bernoulli distributions (target and proposal, respectively). Let $\tilde{P}$ denote the resulting distribution after rejection sampling, and assume a bound on the model's bias $|e(x)| \leq c \in \mathbb{R}$. Then, with $M$ defined in Section 5.1,*

$$\text{TV}(\tilde{P}, P) \leq \frac{Mc}{1 - Mc}. \tag{2}$$

From empirical observations, particularly in Figure 8(b;c), we note that the LLM appears well calibrated when $A(x) = 1$. Therefore, we can further derive the following result.

**Proposition 2** (Informal, see Proposition 2 in Appendix A.4.)**.** *Following Proposition 1, with the additional assumption that $e(x) = 0$ if $A(x) = 1$, i.e., we have $\tilde{A}(x) = A(x) + e(x)$ if $A(x) < 1$, and $\tilde{A}(x) = A(x)$ if $A(x) = 1$. Then, with $\hat{x}$ being chosen such that $A(\hat{x}) < 1$,*

$$\text{TV}(\tilde{P}, P) \leq \frac{Q(\hat{x})Mc}{(1 - Q(\hat{x})Mc)}. \tag{3}$$

This gives a bound for the TV distance between the resulting distribution from VRS ($\tilde{P}$) and the ideal target ($P$). Now, we want to see when VRS is better than direct sampling. We can denote the resulting distribution from direct sampling as $\bar{P}$, and assume it has the same sampling bias $e(x)$. Then, VRS is better than direct sampling if $\text{TV}(\tilde{P}, P) < \text{TV}(\bar{P}, P)$. We can derive the following.

**Corollary 1** (Informal, see Corollary 1 in Appendix A.5.)**.** *Following Proposition 2, and assuming that $\bar{P}$ has the same bias as $\tilde{A}(x)$, i.e., $\bar{P}(x) = P(x) + e(x)$. Then,*

$$\text{TV}(\tilde{P}, P) < \text{TV}(\bar{P}, P) \iff Q(\hat{x}) < \frac{1}{M(1 + c)}. \tag{4}$$

Intuitively, the corollary says that if the proposal $Q$ puts little enough mass, i.e., less than $1/(M(1 + c))$, on the state that sometimes gets rejected ($\hat{x}$), the error introduced inside the rejection step hurts less than applying the same error directly to every draw from $\bar{P}$. In our experiments we fix the proposal to $q = 0.5$. This allow us, for each target $p$, to compute the corresponding $M$ and derive the maximum allowable bias $c$ under which VRS still outperforms direct sampling. In Figure 10, we visualize this by shading the box defined by $\text{clip}(p \pm c, 0.0, 1.0)$ on the top of the calibration plot Figure 4(a). The result shows that in most cases, the empirical frequencies from

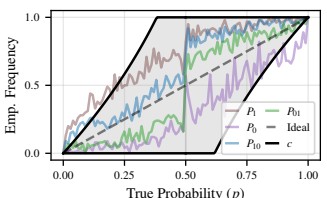

Figure 10: Calibration plots with error bounds $\pm c$ overlaid.

direct sampling fall well within this box, satisfying the theoretical condition. This provides strong evidence that the primary source of VRS's improvement comes from the rejection sampling algorithm itself, not just prompt effects.

# 7 Conclusion and Limitation

We examined the ability of LLMs to sample from natural-language-described distributions, using Bernoulli as a test case. While LLMs can evaluate whether data matches a distribution, they struggle to generate unbiased samples, revealing a clear knowledge–sampling gap. This highlights that *sampling is a fundamentally distinct ability from question answering*: evaluation tasks have clear supervision, while i.i.d. sampling lacks per-instance ground truth and is only verifiable at the distribution level. We tested whether prompt phrasing or chain-of-thought reasoning could reduce bias. While both influence behavior, neither reliably closes the gap. To address this, we proposed Verbalized Rejection Sampling (VRS), a lightweight adaptation of classical rejection sampling expressed entirely in natural language. VRS improves calibration across models without accessing logits or tuning decoding parameters, and our analysis shows that the algorithm, not just prompt design, is key to its success. Beyond correcting this specific failure mode, our work points to a broader path: integrating principled randomness into LLM-based systems. Faithful Bernoulli sampling is a basic requirement for LLM-driven simulations and probabilistic reasoning. VRS illustrates how probabilistic tools can be verbalized and paired with LLMs to improve reliability—without resorting to opaque prompt engineering. This study is limited to the Bernoulli case; our theoretical results do not directly generalize to more complex distributions. Extending this framework to broader families remains an important direction for future work.

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
