# OpenReview forum: "Flipping Against All Odds: Reducing LLM Coin Flip Bias via Verbalized Rejection Sampling"
_NeurIPS.cc/2025/Conference — Submitted to NeurIPS 2025_

### Official Review · Reviewer_rdLt · 2025-06-03

**Clarity:** 3
**Significance:** 3
**Originality:** 3
**Rating:** 5
**Confidence:** 4

**Summary:**

The authors show that rejection sampling can be incorporated into an LLM to improve that LLM’s ability to sample from a Bernoulli. They have some empirical and theoretical evidence to help explain why this approach improves upon direct sampling. The argument is that the algorithm helps correct for some amount of bias present within the LLM whereas that bias is otherwise reflected directly in the outputs under direct sampling.

**Questions:**

0. In figure 8 you lay out the empirical acceptance rates. It appears for p=0.5 the acceptance rate is 1. We can see in figure 7 that the empirical frequency is about 0.5. This is somewhat surprising given the poor results shown in previous work (or say in figure 4.) It seems that the VRS does not empirically impact the model outputs at p=0.5 yet the performance appears much better at that value compared to direct sampling. Can you help me understand what’s going on here?
1. In your conclusion you state that your work points towards a broader path of integrating principled randomness into LLM-based systems. How extensible do you view this direction to be? It seems challenging to use your approach for questions/tasks when these capabilities are part of a larger or complex task.
2. Where does the sample from Q come from? Is it programmatically sampled?
	* If it is sampled by an LLM, how much does the P matter when sampling the value? Would VRS work irrespective of how the candidate was actually sampled?
	* If it is sampled programmatically, does this somehow weaken the result? Basically, if this is the case, I think it needs to be stressed that it requires a random sampler to work.

**Ethical Concerns:**

["NO or VERY MINOR ethics concerns only"]

**Final Justification:**

The paper seems well-done and the authors answered all the questions that I had. Moreover, when I read their overall rebuttal I found it convincing!

**Limitations:**

yes

**Paper Formatting Concerns:**

NO paper formatting concerns

**Quality:**

4

**Strengths And Weaknesses:**

# Strengths

* Overall good presentation, examples, figures, and writing
* Experiments were generally well setup and explained
* Section 6.2 was a nice combination of theory and empirical evidence

# Weaknesses

0. Making/highlighting which parts of VRS are executed by a sampler, by a LLM, by an algorithm would be great. Section 5.2 seems underspecified: “the proposed sample x” – where is this sample coming from?
1. Downstream use of this approach remains unclear (but exciting!)

# Notes
* In figure 2, making clear what part is templated versus an example (0.0, 0.5 values) would help.
* It’s not clear to me why Q was set to 0.5; does 0.5 in particular help?
* Is it possible to re-run the experiments with additional trials to capture the variance; for example, in section 6, table 2, there are no standard errors? It seems possible that the differences between adding the corrected M (versus not) in the prompt could be within noise.

---

> ### Author Rebuttal · Authors · 2025-07-31
>
> We thank the reviewers for their thoughtful and constructive feedback, as well as their recognition of our vision! We are encouraged by the positive evaluations and are grateful for the specific suggestions that will help us improve the clarity and impact of the paper. In the following, we address the main questions and concerns.
>
> ---
>
> > **Where does the sample from Q come from?**
>
> R0: In our implementation, the proposed sample x∼Q (where Q=Bern(0.5)) is generated programmatically. This can be done using standard libraries (e.g., Python’s random module) or deterministically, for example by submitting half the prompts with x=1 and the other half with x=0. To clarify the execution flow, we will add a pseudo-code block in the appendix that explicitly specifies which parts of the VRS loop are handled by a sampler (e.g., sampling x∼Q) and which by the LLM (e.g., deciding to accept or reject x based on the natural-language prompt).
>
> The natural follow-up question is: “Does using a programmatic sampler for Q weaken the result?” This is a very good question we also thought about during the project! Our conclusion is that it does not weaken the results for the following reasons:
>
> * **The LLM’s stochastic behavior is still central.** A crucial step in our method is whether the LLM can reliably carry out the accept/reject decision in a probabilistic way, purely through reasoning over language. This is precisely where LLMs have struggled in direct sampling, and where VRS shows a surprising improvement. The fact that the input x is sampled externally does not diminish this core finding.
>
> * **Programmatic randomness is standard in computational sampling.** Virtually all stochastic processes in simulations or machine learning, whether it's sampling from a Gaussian, Bernoulli, or any complex distribution, ultimately rely on deterministic procedures to generate pseudo-randomness. For example, diffusion models begin with noise sampled from a programmatic Gaussian, which is then transformed into structured outputs (e.g., images).
>
> * **VRS mirrors this classical setup.** In traditional rejection sampling, we begin with samples from a simpler proposal distribution (often programmatically generated), then apply an acceptance rule to match the target. VRS follows this paradigm: x∼Q comes from a simple source, while the LLM plays the key role of evaluating and filtering these proposals to better approximate P.
>
>
> In short, while VRS relies on a basic external sampler for proposals (as do many probabilistic systems), it is the LLM’s ability to perform probabilistic filtering in natural language that enables improved calibration.
>
> We appreciate the opportunity to clarify this point and will make these design choices more explicit in both Section 5.2 and the appendix.
>
> ---
>
> > **It’s not clear to me why Q was set to 0.5; does 0.5 in particular help?**
>
> R1: Thank you for this insightful question! While we initially selected Q=Bern(0.5) for its simplicity and symmetry, we also explored alternative values for q, such as 0.75 and 0.23. Empirically, we found that these alternative settings consistently led to worse calibration in VRS.
>
> Interestingly, the reason for this degradation can be explained with our theoretical analysis in Corollary 1 (Equation 4). This corollary characterizes when VRS yields lower total variation error than direct sampling. We can simply plug-in different values of q to calculate the corresponding c.
>
> In response to your question, we re-ran the analysis behind Figure 10 using q=0.75 and q=0.23. While we are unable to include new plots during the rebuttal phase, the summary is as follows:
>
> * For q=0.75, the shaded "tolerable error" region of Figure 10 shrinks to roughly a quarter of its area and shifts to the right, favoring target distributions P with high probability mass on 1.
>
> * For q=0.23, the region again shrinks and shifts to the left, favoring low-probability targets.
> In both cases, this leads to significantly narrower regions where VRS is guaranteed to outperform direct sampling.
>
> We will add this extended discussion and the additional theoretical analysis to the appendix in the revised version.
>
> ---
>
> > **Is it possible to re-run the experiments with additional trials to capture the variance?**
>
> R2: Absolutely! We have already begun re-running key experiments. Due to computational constraints, these jobs are currently in queue and may take some time to complete. Preliminary results so far indicate that the observed behaviors are consistent. We will create a new comment and revise the paper once they are finished.
>
> ---
>
> > **What’s going on when p=0.5?**
>
> R3: This is a great observation! At p=0.5, the target distribution P exactly matches the proposal distribution Q=Bern(0.5). In classical rejection sampling, this implies that the acceptance probability is 1 for both A(x=1) and A(x=0) (see Eq.(1)), meaning all proposed samples should be accepted. Therefore, the empirical acceptance rate being 1 in Figure 8 is expected and correct.
>
> What makes this case interesting is that the LLM is able to identify this scenario and derive that the acceptance probability is 1, i.e., it is _not required to perform any filtering_ under VRS when p=0.5. Since every sample is accepted, VRS acts as a transparent pass-through for the proposal samples, which are unbiased by construction (i.e., programmatically generated as 50% 0s and 50% 1s). This explains why the empirical output distribution in Figure 7 aligns closely with the true distribution, even if the LLM's internal sampling remains biased in the direct sampling setup (as seen in Figure 4).
>
> In short, the improved performance at p=0.5 under VRS arises not from any correction performed by the LLM, but because no correction is needed, i.e., the proposed samples already follow the target distribution, and the LLM is able to correctly reason through this case and accept all of them.
>
> ---
>
> > **How extensible is this approach or direction?**
>
> R4: Thank you for this important and forward-looking question. We agree that integrating principled randomness into larger LLM-based systems presents both opportunities and challenges. While our current implementation of VRS is evaluated only on Bernoulli distributions, we see it as a foundational step in a broader direction.
>
> * _Conceptual generality._
> Like classical rejection sampling, VRS is conceptually general; it defines a modular sampling procedure that accepts or rejects proposed samples based on the target distribution. In classical setups, adapting rejection sampling to different distributions requires recalculating the acceptance probability and ensuring proper bounding constants. VRS is no different in this respect: the Bernoulli case serves as a minimal yet complete example to walk through the full adaptation process: verbalizing the algorithm, mapping classical theory to natural language setup, benchmarking LLM behavior, and tightening theoretical bounds using empirical insights. While extending VRS to other distributions (e.g., categorical or Gaussian) is non-trivial, the structure of the method is general and can be adapted with further work.
>
> * _General principle: verbalizing probabilistic logic._
> More broadly, the core idea behind VRS, i.e., probabilistic reasoning in natural language using LLMs, is highly extensible. One exciting direction we are exploring is building verbalized generative models, where we can learn a function parameterized by natural language descriptions that maps samples from a simple proposal distribution (e.g., uniform or Gaussian) to more complex target distributions (defined by the training dataset). Such verbalized learning can be done with recent works like Verbalized Machine Learning (VML) [18]. This will open the door to specifying distributions in terms of semantic concepts rather than numeric concepts.
>
> * _Beyond explicit distributions._
> Currently, VRS assumes that the target distribution is explicitly specified. A compelling future direction is to extend this framework to implicitly defined distributions, where we don't have an analytic form of P(x), but perhaps only access to samples or a semantic description. In this setting, we may need to invent new verbalized algorithms, more distant from classical rejection sampling, but still grounded in the same principle. One motivating observation (see Section 4.1) is that LLMs, while poor at directly generating faithful samples, are better at discriminating whether a sample conforms to a target distribution. This discriminator-like capability could form the basis for novel sampling methods, perhaps analogous to how GANs use adversarial feedback for generative modeling.
>
> In summary, we view VRS not just as a standalone technique, but as an instance of a broader strategy: using natural language as a medium for expressing and executing probabilistic algorithms. We outline some possible concrete directions above, we believe these ideas can scale to more complex distributions, larger reasoning chains, and structured downstream tasks. We’ll expand this vision and potential directions in the conclusion of the revised version.

---

> > ### Comment · Area_Chair_BMKj · 2025-08-07
> > **Reviewer: please respond to authors.**
> >
> > Dear Reviewer,
> >
> > Your active participation in the review process is crucial. Please respond to the authors' response and acknowledge you have done so.
> >
> > Thanks.
> >
> > -AC

---

> ### Author Response · Authors · 2025-08-05
>
> [Removed]

---

> > ### Comment · Area_Chair_BMKj · 2025-08-07
> > **Do not post additional rebuttal material after the deadline**
> >
> > Authors: you should not be posting additional rebuttal material after the July 30 deadline by using the "Comment" feature. This is against NeurIPS guidelines.
> >
> > Reviewers: please disregard the contents of the comment made on August 5.

---

> > > ### Author Response · Authors · 2025-08-07
> > >
> > > Dear AC,
> > >
> > > Thank you for your reminder, we have removed the additional material to align with NeurIPS guidelines.
> > >
> > > -Authors

---

### Official Review · Reviewer_oY4x · 2025-06-26

**Clarity:** 4
**Significance:** 2
**Originality:** 2
**Rating:** 5
**Confidence:** 4

**Summary:**

The paper investigates the ability of LLMs to perform sampling in the context of Bernoulli distributions.
It shows empirically that there is a gap between the ability of LLMs to identify a distribution and sample from the same distribution showing that LLMs in general are noisy samplers. The authors then introduce Verbalized Rejection Sampling (VRS) -- a natural language template prompting the LLM to perform rejection sampling -- and show theoretically and empirically that VRS reduces the variational distance (TV distance) between the empirical distribution (induced by the samples) and the target distribution compared to directly prompting the LLM for samples (under some constraints on the sampling error/bias).

**Questions:**

- It was unclear to me if each sample was gathered using a different session. If yes, did you try to ask for multiple samples in the same session and did this affect the sampling error? For example, in VRS if a sample was rejected one could keep asking in the same session for a new sample until one was accepted.
- Did you also compare if the probabilities of each outcome {0,1} in the next-token distribution of the LLM? Did it vary a lot for each sample?

Minor comments:
- Please use "\citet" in the related work section when a paper is a subject in the sentence.
- Section title 6.1: Is it supposed to be "Is the Magic in the Prompt?"
- I think the referenced Figure in line 266 should be Figure 9 a) not Figure 8 c)
- It is not immediately clear what $\hat{x}$ refers to in Equation 4

**Ethical Concerns:**

["NO or VERY MINOR ethics concerns only"]

**Final Justification:**

Although I am skeptical about the practical use of LLM as samplers, I think this paper is a good contribution towards understanding the probabilistic reasoning capabilities of LLMs and should be accepted.

**Limitations:**

To me a very clear limitation of using LLMs as a sampler is that there is no guarantee of any form on the resulting samples.
As the authors show, various aspects such as the model architecture, prompt and seed can influence the sampling error of the LLMs.
In contrast, pseudorandom number generators used for sampling have clear guarantees that make them reliable for the use in downstream tasks. Hence, I think the authors should emphasize that, due to these sampling errors, using LLMs as samplers can have adverse consequences to downstream tasks.

**Paper Formatting Concerns:**

No concerns, but some plots are very small.

**Quality:**

3

**Strengths And Weaknesses:**

**Strengths**

- The paper is well written and easy to follow. The conducted experiments are appropriate and well motivated when presented. Some details could have been clarified better (see questions).
- The theoretical results shown in section 6.2 were especially insightful and added a lot to the understanding of why their method VRS works better than directly asking the LLM for samples (also the result is more general and does not only apply to LLMs but any noisy sampler.)

**Weaknesses**

- I think the work is conceptually interesting, but I don't see the practical use/significance of LLMs as samplers. While the authors motivate it as a first step towards more advanced probabilistic algorithms (Monte Carlo algorithms, randomized decision making, agent-based simulations, etc.), usually the theory behind these algorithms assumes true in distribution samples and, thus, any kind of noisy sampler (with unknown error) would lead to unexpected/unreliable results. (And as the authors show, there are many factors such as prompting and model architecture that influence this error.) If we want LLMs to be able to run advanced probabilistic algorithms, making the LLM use a pseudorandom number generator as a tool seems more reasonable to me.
- The presented methodology is limited to Bernoulli distributions

---

> ### Author Rebuttal · Authors · 2025-07-31
>
> We thank the reviewer for the thoughtful and encouraging feedback. We appreciate the recognition of the clarity of the writing, the soundness of our theoretical analysis, and the value of the empirical study. Below, we first address the questions, followed by discussion of the concerns.
>
> ---
>
> > **Are samples gathered using different sessions? Did you try sampling within the same session?**
>
> R1: Yes, all samples were gathered via stateless API calls. This ensures that the samples are i.i.d.
>
> That said, we recognize that the reviewer’s question touches on a different and equally interesting regime, closer to sampling with memory or conditioning. In response, and out of curiosity, **we ran additional experiments with stateful VRS**, where rejected samples are followed by resampling within the same session. Interestingly, we found no noticeable difference in the sampling error or the acceptance rate compared to the stateless baseline. A likely reason is that our prompt explicitly instructs the model to make an i.i.d. decision, and we do not explicitly instruct the model to use prior history.
>
> It would be possible to design non-i.i.d. samplers where the prompt explicitly references and instructs the model to reason over past samples. **To explore this further, we also additionally ran a variant of direct sampling** where the sampling is done one by one and the prompt includes all previously sampled values with explicit instructions to make use of them. This setup improved over i.i.d. direct sampling but still underperformed compared to VRS. While a full analysis is beyond our current scope, we agree this opens a compelling direction for modeling sequential stochastic behavior.
>
> ---
>
> > **Did you compare probabilities for {0,1} in the LLM’s next-token distribution?**
>
> R2: Thank you for the question. While our main setup assumes a black-box interaction with the LLM, where internal outputs like logits are inaccessible, **we conducted additional experiments** for this question to examine next-token probabilities for {0,1} in direct sampling, using both CoT and non-CoT prompts with the prompt phrasing P1 in Figure 4..
>
> We found that the token probabilities vary significantly across samples and target probabilities, particularly in the non-CoT setting. For example, at p=0.5, the model without CoT assigns ~96% probability to “1”, indicating strong bias which is consistent with the empirical frequency shown in Figure 4(a). Moreover, we observed high variance in token-level probabilities across samples, especially near p=0.0 and p=1.0, despite identical outputs.
>
> In contrast, CoT prompts yield more stable and sharply bimodal token distributions. For example, at p=0.3, CoT prompting results in a clear separation: when the samples are "1", the token probabilities are P(1) ≈ 0.93, while for samples "0" the token probabilities are P(1) ≈ 0.14. One explanation is that the reasoning in CoT will increase the probability to a specific outcome, the last token at the end will almost surely be this outcome. This is also noted in our maintext (line 115).
>
>
> ---
>
> > **Minor Comments and Edits**
>
> R3: Thank you for these pointers. We will:
> * Update references to use \citet where appropriate in related work.
> * Change “Is it the Magic in the Prompt?” to “Is the Magic in the Prompt?” in the section title.
> * Actually, it is indeed Figure 8(c) for line 266, which shows both the analytical and empirical acceptance probability, i.e., A(1) and $\tilde{A}(1)$. In line 274, we refer and describe Figure 9(a) for the first time, which is a calibration plot.
> * Rephrase the corollary 1 to make it clearer that $\hat{x}$ refers to those that $A(\hat{x})<1$
>
> ---
>
> > **Practical Significance of LLMs as Samplers**
>
> R4: This is a very insightful comment, and we appreciate the opportunity to elaborate. As the reviewer notes, our work represents an early step toward enabling and understanding more advanced verbalized probabilistic algorithms. The setup in this paper, focused on Bernoulli distributions, is intentionally simple, not because the problem is trivial, but because it offers a concrete foundation to study a deep and emerging capability: probabilistic reasoning in natural language.
>
> We fully agree that in classical settings, sampling should rely on tools, which offer well-defined guarantees. However, our motivation arises from realistic and increasingly common LLM deployment scenarios where: LLMs act as autonomous _agents_ expected to make decisions involving chance (e.g., tie-breaking, randomized planning), interfaces are purely natural language, with no tool execution available or permitted, even when tools are available, their invocation may compromise interpretability, modularity, or security (e.g., sandboxed educational or fairness-sensitive settings).
>
> While invoking an external tool is technically sound, relying on tool use as a universal solution may not be realistic or sufficient. Many LLM-based systems already operate in tool-free settings, and users (often unknowingly) trust the LLM’s verbal reasoning to simulate stochasticity. We view VRS not as a replacement for principled samplers, but as a practical, language-native safeguard that significantly reduces sampling error in such environments.
>
> In the long term, this raises several foundational questions: How can we understand and control stochastic behavior in LLMs through reasoning? What are the algorithmic abstractions that can be embedded within language? How robust are these probabilistic reasoning? These are open and important challenges. As the reviewer correctly notes, LLMs currently do not offer distributional guarantees. But if we want to reason about and improve their probabilistic reasoning capabilities, we must begin somewhere, and our work aims to provide that conceptual and empirical example.
>
> Finally, while VRS currently assumes access to an explicit target distribution P(x), a compelling future direction is to extend this framework to implicitly defined distributions, where P(x) is only described semantically or via examples, rather than analytically. In such cases, tool use may no longer help, as there is no closed-form function to evaluate. Interestingly, our findings in Section 4.1 show that LLMs are often better at recognizing whether a sample fits a distribution than at generating it. This discriminator-like ability could inspire new verbalized sampling paradigms, perhaps analogous to adversarial models like GANs, where judgment about sample quality is used to refine generative behavior.
>
> In short, we agree that faithful sampling from LLMs is a difficult and unresolved problem, but it is precisely because it is difficult, and increasingly relevant, that we believe it deserves attention now.
>
> ---
>
> > **The presented methodology is limited to Bernoulli distributions**
>
> R5: **While our evaluation focuses on Bernoulli distributions, we want to emphasize that the limitation does not lie in the generality of the methodology.** We view this work as a proof of concept for a broader class of natural-language-based probabilistic methods, where classical probabilistic algorithms can be verbalized and executed using LLMs in black-box settings. Bernoulli distributions were chosen not because they are the end goal, but because they are the simplest nontrivial setting that allows for clean isolation and analysis of the fundamental questions. The simplicity of the Bernoulli setting enables us to walk through the full adaptation pipeline (i.e., prompt construction, theoretical framing, empirical benchmarking) and establish core insights that would otherwise be obscured in higher-dimensional setups.
>
> Conceptually, VRS, like classical rejection sampling, is structurally general. Adapting it to other distributions, such as categorical, binomial, or even continuous ones, will require customized derivations of the different performance bound and empirical analysis, just as in the classical case one will need to derive a new acceptance probability for different target or proposal distributions. But the overall framework does transfer: i.e., verbalizing the algorithm, mapping classical theory to natural language setup, benchmarking LLM behavior, and tightening theoretical bounds using empirical insights. **Inspired by reviewer feedback, we extended our theoretical results** and showed that Propositions 1 and 2, as well as Corollary 1, generalize to Binomial distributions and to other discrete distributions under mild assumptions (e.g., that the LLM behaves correctly when the acceptance probability is 1). We will include these generalized results in the updated version. However, practically deploying VRS on richer distributions will require deeper investigation.
>
> In short, while the current evaluations are limited to Bernoulli sampling, the methodology, theoretical framework, and core insights are general, and we see this work as the beginning of a broader research direction in language-native probabilistic reasoning. We will revise the paper to better highlight these generalizations, include the extended theoretical results, and clarify the distinction between the evaluated scope and the structural potential of VRS.
>
> ---
>
> > **Limitations: Emphasizing Risks of Using LLMs as Samplers**
>
> R6: We agree with the reviewer’s final comment. LLM-based sampling lacks the guarantees of standard random number generators. This is a core motivation of our work: to highlight these risks and offer a structured way to mitigate them (via VRS). We will revise the Limitations section to explicitly reinforce that uncontrolled use of LLM sampling can lead to bias, and that VRS is only a partial remedy.

---

> ### Comment · Reviewer_oY4x · 2025-08-05
>
> I thank the authors for the clarifications. I think the example the authors gave of future directions where P(x) is only described semantically or via examples is quite interesting. Personally, I am still skeptical though if, more generally speaking, such verbalized probabilistic algorithms should be used in practice as it might obfuscate (modeling) assumptions made by the LLM and the errors in the generated outcomes. However, I agree with the authors that we should aim to understand the probabilistic reasoning capabilities of LLMs and this paper is a good example towards this goal.
>
> Something that is confusing though, is that the authors write that tool usage might not be possible due to, e.g., modularity or security issues, but the VRS prompt does utilize an external sampler, e.g., Python’s random module, for the proposed sample $x \sim Q$ (as described in the answer to Reviewer rdLt).

---

> ### Author Response · Authors · 2025-08-05
>
> Thank you for the follow-up! We appreciate your continued engagement and your acknowledgment that understanding the probabilistic reasoning capabilities of LLMs is an important goal.
>
> Regarding the point on tool usage, we agree that this deserves clarification. When we mention that tool usage may be infeasible, we are specifically referring to **_LLM-initiated tool execution_**, where the model generates code and then runs it autonomously, often in a sandboxed environment (e.g., as with code execution tools in ChatGPT, where the code is executed on OpenAI’s servers). Such sandboxing may not always be available and often requires nontrivial system setup. More importantly, it raises security and reliability concerns, especially in production or safety-critical contexts, where letting the LLM autonomously execute arbitrary code may not be acceptable.
>
> In contrast, the execution of $x \sim Q$ in VRS is done by the user in the known and controlled environment. In VRS, the LLM itself does not generate or execute code; it simply receives the sampled value as input and performs the accept/reject decision in natural language. This preserves full user control over the sampling process, avoids the risks of autonomous execution, and keeps the model operating within a transparent, text-only interface.
>
> We’ll clarify this distinction when we revise the paper, and we appreciate the reviewer for highlighting the opportunity to make it more explicit.

---

### Official Review · Reviewer_VD8B · 2025-07-03

**Clarity:** 3
**Significance:** 2
**Originality:** 2
**Rating:** 4
**Confidence:** 4

**Summary:**

In this work, authors observe that asking an LLM for samples from a parameterized distribution is often biased, even when that model has the capability to correctly classify parameters from a set of true samples. The authors propose an approximation of a rejection sampling method that uses direct responses from the LLM as the reject/accept step. Empirically they find this reduces calibration error when sampling from a Bernoulli distribution.

**Questions:**

* Have the authors explored using alternative MCMC algorithms to improve efficiency of their sampling procedure?

**Ethical Concerns:**

["NO or VERY MINOR ethics concerns only"]

**Final Justification:**

My main concern is whether this method can generalize beyond the Bernoulli distribution. I think the authors have helped to alleviate this very slightly with their claim of an extension of the theory to Binomial distributions, where at least one can get a better sense of what acceptance error bound one would need so that the overall bound isn't vacuous. There is still the issue of unrealistic assumptions and computational costs, which I don't think can be validated without further experiments. Despite this, the originality of the research question makes it so that I think the paper might still serve as a useful starting point for future research into this topic.

**Limitations:**

Yes

**Quality:**

3

**Strengths And Weaknesses:**

Strengths
* The high level research question, I.e. can we make use of the distributions implicitly learned by LLMs, I believe is an extremely useful one. The authors argument of preventing biased responses from a user-safety perspective I think is valid as well
* Sections 4.1 , 4.2 provide a clear example of miscalibration and motivates the method

Weaknesses
* The method presented is heuristic, which I believe would be okay if validated on more complex distributions (e.g. even Binomial) for a range of hyper parameter settings such as temperature. As it stands I am not convinced that this method necessarily translates well to more complex settings, especially since it appears fairly sensitive to the rejection prompt (paragraph starting on line 290).
* This method seems extremely computationally intensive, especially since each rejection step uses chain of thought reasoning. Other MC sampling methods exist which are sometimes more efficient (MCMC, Langevin, HMC) but authors do not explore analogous extensions
* The two points above lead to my main concern about the practical use case for this method. Why not trigger the LLM to call an external sampler for these simple distributions? If the model has secretly learned more complex distributions (e.g. the number of times the word “dog” appears in news articles) how can we efficiently sample and validate that VRS still works here? I think further experiments are needed to answer these questions.

---

> ### Author Rebuttal · Authors · 2025-07-31
>
> We thank the reviewer for their feedback and recognition of the high-level vision behind our work. Below we address the main concerns raised.
>
> ---
>
> > **The method is heuristic, does it generalize beyond Bernoulli?**
>
> R1: Good question! In fact, **VRS is a prime example of how prompt-based LLM methods can go beyond heuristics.** While much of prompt engineering relies on empirical tuning or intuition, our approach is fundamentally different: we construct a prompt that instantiates a well-established, theoretically grounded algorithm, i.e., rejection sampling, within the language interface of an LLM.
>
> VRS is not inspired by heuristics; it is rejection sampling adapted to natural language. More importantly, we go beyond simply borrowing a classical idea: in Section 6 and Appendix A, we provide a theoretical analysis of VRS in the LLM setting. We derive explicit conditions under which VRS improves over direct sampling, even when the LLM is biased. These guarantees generalize to any noisy sampler (as _also noted by Reviewer oY4x_), and demonstrate that the performance gains of VRS are not empirical artifacts, they follow from provable properties of the algorithm itself. In this sense, VRS is theory-backed, not heuristic-driven.
>
> Regarding generalization beyond Bernoulli: we focused on Bernoulli distributions due to their interpretability and tractability, which allow for fine-grained analysis of stochastic behavior in LLMs. We view this work as a proof of concept for a broader class of natural-language-based probabilistic algorithms. The methodology, the construction of algorithmic prompts, the embedding of classical stochastic procedures in natural language, and the theoretical analysis of bias propagation, generalizes in structure even though each distribution will require its own adaptation.
>
> The VRS framework, like classical rejection sampling, can in principle be extended to any distribution where distributions can be expressed textually. We do not claim that our specific Bernoulli prompt generalizes; rather, we emphasize that the framework of verbalizing probabilistic tools inside LLMs, and analyzing them with classical theory, is general.
>
> **Inspired by the reviews, we are now able to extend Proposition 1 and Proposition 2 and show that they generalize to Binomial distributions** (as well as other distributions with discrete state spaces, same distributional form of P and Q, and a single outcome that maximizes the acceptance ratio) under the assumption that the LLM arrives at the right decision if the acceptance ratio is 1 (which we observe to be true for the Bernoulli case). Additionally, the bound presented in Corollary 1 also generalizes under the assumption that direct sampling yields a distribution within TV distance $c$ of the target. We will update the paper with the new theoretical results.
>
> Making statements about the practical performance of VRS would (a) require us to tighten the analysis about the error of direct sampling and (b) investigate the tradeoff involving the probability $\sum_{k: A(k)<1} Q(k)$. Both (a) and (b) are crucially dependent on the distribution at hand. Additionally, (a) is an empirical question dependent on the LLM and its sampling behavior. We note that none of these arguments invalidates VRS but opens opportunities for future research.
>
> Lastly, the paragraph starting at line 290, which discusses prompt effectiveness, has perhaps been misinterpreted. Our goal there is not to suggest reliance on the prompt; rather, the opposite: we show that even imperfect prompts can still benefit from VRS’s algorithmic structure. The performance gains do not hinge on perfect prompt design. This reinforces our broader point: it is the probabilistic algorithm embedded in the prompt, not the wording itself, that drives the improvement.
>
> In sum, VRS represents a shift away from ad hoc prompt heuristics and toward algorithmically grounded, analyzable, and extensible methods for improving the reliability of LLM-based sampling. We believe this makes it an important step toward robust probabilistic reasoning with LLMs.
>
> ---
>
> > **The method is computationally intensive.**
>
> R2: We agree that VRS incurs higher computational cost than direct sampling, and this is both expected and meaningful. The key point, however, is that VRS is an algorithm that operates in a fundamentally different space: the informal, natural language domain.
>
> In classical settings, sampling algorithms like rejection sampling or MCMC are implemented in formal programming environments (e.g., Python). These are efficient, but they also assume the user has already formalized the problem, encoded it in code, and specified exact parameters (e.g., manually calculated the acceptance probability). In contrast, VRS addresses a very different use case: the user specifies the problem informally in natural language, and the computation itself happens within the language space.
>
> This shift, what we might call _verbalized computing_, has important implications. While inference in this space may be more computationally expensive, it is also more accessible. A user can invoke VRS by simply describing a desired distribution in plain text. There is no need to write or call code, craft sampling functions, or define rejection logic programmatically. This convenience is not free, but it lowers the barrier of entry for users who otherwise would not engage with formal stochastic computation. Viewed this way, **“cost” of VRS is offset by the elimination of the cost of formalization, which is often unacknowledged in computational models, yet a dominant factor in practice.**
>
> Moreover, _we believe this reframes how we think about computational complexity in the LLM era._ Traditional complexity theory does not account for the cost of formalization, and directly starts with an already formalized problem to analyse its complexity. In LLM-based systems, where both problem specification and computation occur in natural language, the relevant complexity includes the effort saved by not formalizing the task, and that’s where natural-language-based algorithms like VRS shines.
>
> We also want to clarify that the actual sampling overhead of VRS is bounded and modest in the Bernoulli case. Since we use a symmetric proposal distribution (q = 0.5), the worst-case acceptance probability is 0.5, meaning we expect to draw twice as many proposals as needed samples in the worst case. This remains in the same complexity class: generating n accepted samples via VRS still takes O(n) calls to the model.
>
> **On MCMC and Future Work.** We’re excited by the reviewer’s mention of MCMC! It is exactly the kind of follow-up we hope to inspire. Verbalizing MCMC introduces new challenges, which we believe deserves dedicated attention: it is inherently stateful and sequential, requiring memory across steps and possibly longer reasoning chains. This would likely necessitate chain-of-thought and multi-turn prompts, increasing per-sample cost. In contrast, VRS is fully stateless and parallelizable, with i.i.d. outputs, making it especially practical in current LLM settings.
>
> A well-designed verbalized MCMC might improve sampling _efficiency_, but may be even more **computationally intensive** than VRS. Its design would merit a paper of its own; we see our work as laying foundational groundwork for such exploration.
>
> ---
>
> > **Why not ask the LLM to call an external sampler?**
>
> R3: We fully agree that when an external tool, e.g., Python-based sampler, is available and callable, it can be used to generate unbiased samples efficiently. However, our work is not positioned as a replacement for such tool-assisted approaches. Instead, we focus on a different and complementary question:
>
> _Can large language models, operating solely in natural language, simulate stochastic processes faithfully without access to external tools or code?_
>
> This question arises from realistic and increasingly common LLM deployment scenarios where: LLMs act as autonomous _agents_ expected to make decisions involving chance (e.g., tie-breaking, randomized planning), interfaces are purely natural language, with no tool execution available or permitted, even when tools are available, their invocation may compromise interpretability, modularity, or security (e.g., sandboxed educational or fairness-sensitive settings).
>
> In such settings, we are left with the LLM itself as the only accessible computational mechanism. The goal of VRS is to explore whether LLMs can simulate stochasticity internally, using only structured prompting. This is not about generating perfect randomness, but about understanding and improving the LLM’s native stochastic behavior, an ability that is underexplored but increasingly relevant as LLMs are deployed as autonomous agents and decision-makers.
>
> More broadly, VRS is not just a sampling method, **it is a case study in how to build and analyze algorithmic prompts in a principled way.** Rather than relying on heuristic prompt engineering, we derive a prompt-based implementation of rejection sampling and provide formal theoretical guarantees for its behavior under model bias. This methodology, i.e., analyzing prompts through the lens of classical algorithms and error bounds, offers a new paradigm for prompt design that bridges empirical performance with formal analysis.
>
> **The analogy here is to research on LLMs’ math capabilities**: although it’s trivial to solve math problems by calling a calculator, we still study whether LLMs can reason through equations in language, because it tells us something fundamental about their internal representations and limitations. In the same spirit, VRS asks whether LLMs can simulate randomness themselves, not by outsourcing it, but by verbalizing and executing probabilistic logic in language.
>
> ---
>
> > **Have the authors explored alternative MCMC algorithms?**
>
> R4: Good question! Please see the second half of the response R2.

---

> > ### Comment · Reviewer_VD8B · 2025-08-08
> > **Response to Rebuttal**
> >
> > Thank you to the authors for their thorough response to each of my concerns. Overall while I am not completely satisfied by some of the computational and generalizability arguments, I think enough context has been added (e.g. binomial theory extension) where I am okay improving my score. My thoughts for specific comments are below.
> >
> > > VRS is not inspired by heuristics; it is rejection sampling adapted to natural language.
> >
> > Thank you for the clarifying thoughts here, and I understand that the bounds derived are valid for rejection sampling in the case of Bernoulli distributions. However as reviewer Q2Ka has noted this theory does not support the claim of VRS being a *generalizable method*. In particular the assumption for proposition 2 that $e(x)=0$ if the true acceptance rate is $1$ is unlikely to hold in more complex scenarios.
> >
> > > we are now able to extend Proposition 1 and Proposition 2 and show that they generalize to Binomial distributions
> >
> > Thank you for considering the extension to binomial distributions, I think stating the propositions in terms of this distribution should be able to give a better sense for how tight the derived bounds are.
> >
> > > even imperfect prompts can still benefit from VRS’s algorithmic structure
> >
> > While this may be technically true, I am concerned that for more complex distributions the improvement may be only very slight and require a large amount of rejected samples unless the right prompt is chosen.
> >
> > > cost” of VRS is offset by the elimination of the cost of formalization
> >
> > The point about formalization cost for complex distributions is a good one. However I think a concrete example where formalization cost for a formal sampler dominates LLM VRS sampling costs is needed (or at least discussed). In practical settings the number of worst case samples is not lower bounded like the Bernoulli case and I would expect this to grow quickly with distribution complexity.
> >
> > >MCMC
> >
> > Thanks for discussing the tradeoffs here
> >
> > > Calling an external sampler
> >
> > Thank you for the discussion on this. The setting where the model may not have the right tools available I agree is realistic, but without stronger theoretical bounds on error or empirical evidence that VRS works on complex distributions, there is still the question of *when to trust* these samples from the model. As mentioned in my initial review I do agree with the authors that uncovering LLMs distributional capabilities is an interesting and useful line of work.

---

> > > ### Author Response · Authors · 2025-08-08
> > >
> > > We thank the reviewer for their follow-up and for improving their score! We especially appreciate the recognition that “uncovering LLMs’ distributional capabilities is an interesting and useful line of work.”
> > >
> > > We agree that questions around computational scalability and generalization to complex distributions are important and warrant deeper exploration. In fact, we see these challenges as natural and exciting next steps building on our work.
> > >
> > > Our goal in this paper was to highlight the overlooked gap between LLMs’ distributional understanding and their sampling behavior, and to offer a concrete, theoretically grounded method as a case study in algorithmic prompting. While we do not claim to resolve the full space of stochastic simulation with LLMs, we believe our contribution provides a rigorous and extensible foundation for future work in this area.
> > >
> > > Thank you again for your constructive feedback and for helping shape the conversation around this emerging topic.

---

> ### Author Response · Authors · 2025-08-05
>
> [Removed]

---

> ### Comment · Area_Chair_BMKj · 2025-08-07
> **Please respond to authors.**
>
> Dear Reviewer,
>
> Your active participation in the review process is crucial. Please respond to the authors' response and acknowledge you have done so.
>
> Thanks.
>
> -AC

---

> ### Comment · Area_Chair_BMKj · 2025-08-07
> **Do not post additional rebuttal material after the deadline**
>
> Authors: you should not be posting additional rebuttal material after the July 30 deadline by using the "Comment" feature. This is against NeurIPS guidelines.
>
> Reviewers: please disregard the contents of the comment made on August 5.

---

> > ### Author Response · Authors · 2025-08-07
> >
> > Dear AC,
> >
> > Thank you for your reminder, we have removed the additional material to align with NeurIPS guidelines.
> >
> > -Authors

---

### Official Review · Reviewer_Q2Ka · 2025-07-04

**Clarity:** 3
**Significance:** 2
**Originality:** 3
**Rating:** 3
**Confidence:** 4

**Summary:**

The paper investigates the discrepancy between Large Language Models' (LLMs) ability to sample from probability distributions and their difficulty in generating unbiased samples from them, focusing on Bernoulli distributions. The authors propose Verbalized Rejection Sampling (VRS), a natural-language adaptation of classical rejection sampling, which prompts LLMs to reason about and accept or reject proposed samples. The paper empirically demonstrates that VRS substantially reduces sampling bias across various LLMs compared to direct sampling. Theoretical analysis is provided, suggesting that VRS improves over direct sampling under certain assumptions, attributing gains to both the algorithm and prompt design. The work aims to show how classical probabilistic tools can be verbalized and integrated into LLM workflows without requiring internal model access or extensive prompt engineering.

**Questions:**

1. Could the authors provide an analysis of whether VRS influences or correlates with the underlying logit probabilities of the LLM? Is there evidence that VRS helps align the LLM's internal logit distribution with the intended Bernoulli distribution? If not then how is it different from other methods.
2. Figure 2: Please clarify why Figure 2 uses "T" and "F" as outcomes instead of the standard "0" and "1" for VRS
3. Could the authors elaborate on whether they consider VRS a novel general method for improving LLM sampling, or rather a specific application of few-shot prompting combined with rejection sampling?

**Ethical Concerns:**

["NO or VERY MINOR ethics concerns only"]

**Final Justification:**

My primary concern remains, the primary claim of the paper is about the following statement, which would make it significant.

I am afraid Prop 1 and 2 are not enough to claim that it as a principaled method, I am not saying that it cannot be, but the empirical justification must be provided, and I encourage the authors to take this forward.

Although, with the reclarifications, and writing changes, I will still update the score.

**Limitations:**

The authors acknowledge in Section 7 that the theoretical analysis is currently limited to Bernoulli distributions. However, the discussion of limitations could be more robust. The paper does not adequately address:

The significant limitation of not comparing VRS against existing and widely used tool-calling methods for generating unbiased samples.

The lack of analysis regarding the LLM's internal logit probabilities and how VRS interacts with or corrects potential biases at that fundamental level.

**Paper Formatting Concerns:**

No issues

**Quality:**

2

**Strengths And Weaknesses:**

Strengths:
1. The paper provides theoretical analysis, including propositions and bounds, which offers some mathematical backing for VRS's improvements, although only for Bernoulli distributions.
2. The ablation studies and analysis are rigorous

Weaknesses:
1. Problem Motivation and Impact: The paper's problem statement lacks real world motivation and clear direction on the practical impact. It fails to demonstrate actual downstream applications where VRS provides a tangible advantage over existing tool based methods (e.g., Python code generation for randomness) which they claim in the introduction.

2. Narrow scope and Generalization: The method, VRS, is explicitly stated not to generalize beyond Bernoulli distributions. The strict limitation to Bernoulli distributions severely curtails the paper's broader significance, and raises more concerns, leaving it to future work does not seem appropriate.

3. Lack of comparisons: The authors deliberately avoid comparisons with tool-calling and logit methods claiming that having LLMs sample from distribution without them will help it in solving ties / survey without any evidence for the latter. The method is analogous to few-shot prompting, where VRS prompts the LLM with examples shown from the same task (Bernoulli sampling) with different parameters (p<1).

---

> ### Author Rebuttal · Authors · 2025-07-31
>
> We thank the reviewer for their feedback and suggestions. We respectfully address what we believe are key misunderstandings about the nature and goals of our work before responding to the remaining points.
>
> ---
>
> > **Clarifying Misunderstanding: VRS is Not a Few-Shot Prompting Method**
>
> R1: We would like to clarify a core misunderstanding: VRS is not a few-shot prompting method, but a structured natural-language implementation of a classical algorithm. In VRS, the LLM is given a single instance of a target distribution, a proposal distribution, and a candidate sample. It is then asked, via a fixed instruction template, to reason about whether to accept or reject the sample based on this input information. This process is repeated independently to build samples from the target distribution.
>
> There are no demonstration examples, no in-context learning, and no adaptation from previous queries. Instead, the LLM is executing a natural-language instruction (accept/reject logic) conditioned on input values, resembling algorithmic reasoning more than imitation of the inputs (also, there is nothing to imitate with). Each invocation of VRS is stateless, self-contained, and purely instructional.
>
> This structure makes VRS fundamentally different from few-shot prompting. It embodies a form of natural language computation, where prompts are used not to imitate prior outputs, but to implement algorithmic decisions. We will clarify this distinction in the paper.
>
> ---
>
> > **Clarifying Motivation and Broader Impact**
>
> R2: Our work is not intended as a replacement for tool-calling methods such as Python-based samplers. Instead, we focus on a different and complementary question:
>
> _Can large language models, operating purely in natural language, simulate stochastic processes faithfully without access to external tools or code?_
>
> We agree that in classical settings, sampling should rely on tools, which offer well-defined guarantees. However, our motivation arises from realistic and increasingly common LLM deployment scenarios where: LLMs act as autonomous _agents_ expected to make decisions involving chance (e.g., tie-breaking, randomized planning), interfaces are purely natural language, with no tool execution available or permitted, even when tools are available, their invocation may compromise interpretability, modularity, or security (e.g., sandboxed educational or fairness-sensitive settings).
>
> In such settings, the LLM itself is the only accessible computational mechanism. VRS explores whether LLMs can simulate stochasticity using structured prompting, i.e., without external randomness or code execution. This is not about outperforming external tools, but about probing the LLM’s _intrinsic_ ability to represent and act on probabilistic logic in language.
>
> More broadly, VRS is not just a sampling method, **it is a case study in how to build and analyze algorithmic prompts in a principled way.** Rather than relying on heuristic prompt engineering, we derive a prompt-based implementation of rejection sampling and provide formal theoretical guarantees for its behavior under model bias. This methodology, i.e., analyzing prompts through the lens of classical algorithms and error bounds, offers a new paradigm for prompt design that bridges empirical performance with formal analysis.
>
> A useful analogy is the study of **LLMs’ math abilities**: although we could trivially solve arithmetic problems via tool calls, we still examine whether LLMs can reason through equations in language, because it reveals something fundamental about their internal representations. In the same spirit, we ask whether LLMs can simulate probabilistic reasoning internally, not by outsourcing it, but by executing probabilistic algorithms expressed in language. We believe this ability is both underexplored and essential for assessing the limits and potential of general-purpose language models. We will clarify this broader motivation in the revised version of the paper.
>
> ---
>
> > **Does VRS influence or align the LLM's internal logits?**
>
> R3: This is an excellent question, and in fact, it gets to the heart of why we believe VRS is both interesting and distinct from other approaches.
>
> **1. VRS does not modify or align logits.**
> Unlike direct sampling, where the LLM is prompted to output "0" or "1" and the resulting logits directly correlate with the sample distribution, VRS prompts the model to sample a decision to accept (T) or reject (F) the proposed sample. Thus, VRS operates over a different output space and does not influence or depend on the logits used in direct sampling.
>
> While VRS does not try to align the model’s internal probabilities, it does yield samples that better match the target distribution. As shown in Section 6.2, this is not due to logit manipulation but to the algorithmic structure imposed by the prompt. This distinguishes VRS from tool-calling (which delegates randomness to external code) and from methods requiring internal model access.
>
> **2. VRS intentionally accepts biased logits and works around them.**
> In contrast to methods that modify LLM behavior by adjusting weights (via fine-tuning) or prompts (via prompt engineering), VRS embraces the fact that the logits are biased and uses a probabilistic mechanism (executed by LLMs) to correct for it, without needing access to or control over the internal distributions. This is conceptually aligned with classical sampling theory: for decades, rejection sampling and similar methods have been used to generate unbiased samples from biased sources. VRS brings this idea to the language interface of LLMs.
>
> ---
>
> > **Why does Figure 2 use "T" and "F" instead of "0" and "1" for VRS?**
>
> R4: Thank you for pointing this out. We use "T" and "F" in the VRS prompts to represent “accept” and “reject” decisions, respectively, not to denote sampled values. This distinguishes the decision made by the LLM (whether to accept a proposed sample) from the sampled value itself, which is either "0" or "1". Using "0" and "1" in both stages could lead to confusion, especially since VRS separates sampling from the decision process. We will clarify this distinction explicitly in the caption and body of the paper to avoid ambiguity.
>
> ---
>
> > **Is VRS a general method or a specific few-shot application?**
>
> R5: VRS is not a few-shot prompting method; please see our clarification in the first part of the rebuttal. It does not rely on in-context examples, but instead instructs the LLM to perform a probabilistic decision based on newly sampled inputs in each query. Every invocation is independent, stateless, and based on a fixed algorithmic template.
>
> To the best of our knowledge, the method is **novel**. While our current work focuses on Bernoulli distributions, we do not claim that VRS, at this stage, solves sampling for arbitrary distributions. However, it demonstrates that a classical probabilistic method, rejection sampling, can be effectively adapted to the natural language domain and executed by LLMs without access to logits or external tools.
>
> **Note that we do not claim VRS does not generalize beyond Bernoulli,** rather, our current empirical studies and theoretical analysis are specific to Bernoulli. In principle, VRS can be extended to other distributions. Just like in classical rejection sampling, each new distribution requires a tailored derivation of acceptance probabilities. Extending VRS will likely require distribution-specific modification and theoretical analysis, which is feasible, but nontrivial.
>
> We view VRS as a **proof of concept for a broader class of natural-language-based probabilistic methods**. It shows that we can: (1) adapt probabilistic reasoning algorithms to LLMs through language, (2) rigorously analyze their behavior using classical tools, and (3) derive formal guarantees even when the underlying sampler (the LLM) is biased or noisy. In this sense, VRS is algorithmically general.
>
> _As Reviewer oY4x noted_, the theoretical analysis we provide in Section 6.2 is itself general and applies to any noisy accept/reject sampler, not just LLMs. **Inspired by the reviews, we were able to show that Proposition 1 and Proposition 2 generalize to other distributions** with discrete state spaces under the assumption that the LLM arrives at the right decision if the acceptance ratio is 1 (which we observe to be true for the Bernoulli case).
>
> Additionally, the bound presented in Corollary 1 also generalizes under the assumption that direct sampling yields a distribution within TV distance $c$ of the target. This is a fairly uniform assumption about the error of direct sampling. For a tighter analysis one would need to investigate the structure of the error of direct sampling case-by-case (i.e., distribution-by-distribution) which we consider out-of-scope for the current work. We will update the paper with the new theoretical results.
>
> ---
>
> > **Limitations: No Tool-Calling Baselines and No Logit-Level Analysis**
>
> R6: We appreciate the reviewer’s suggestions regarding the limitations section. As noted in our clarification, our goal is not to outperform tool-calling or logit-access methods, but to explore whether LLMs can simulate stochastic processes purely via natural language, without relying on external code or model internals. Tool-calling approaches test an LLM’s coding ability, while VRS tests its ability to reason probabilistically in language, a different and underexplored capability.
>
> Similarly, while we do not analyze how VRS affects token-level logits, this is intentional: our framework treats LLMs as black-box samplers. Understanding how logits behave under VRS prompting could be interesting for future work, but it lies outside our current focus on prompt-level behavior.
>
> We agree that these assumptions and trade-offs should be made more explicit, and we will revise the limitations section to make them clear.

---

> > ### Comment · Reviewer_Q2Ka · 2025-08-07
> > **Thanks for your detailed rebuttal**
> >
> > Thanks for the detailed rebuttal
> >
> > > We will clarify this distinction explicitly in the caption and body of the paper to avoid ambiguity.
> > > We agree that these assumptions and trade-offs should be made more explicit, and we will revise the limitations section to make them clear.
> > > There are no demonstration examples, no in-context learning
> >
> > Thanks for these, I have a better understanding now, however my primary concern remains, the primary claim of the paper is about the following statement, which would make it significant.
> >
> > > More broadly, VRS is not just a sampling method, it is a case study in how to build and analyze algorithmic prompts in a principled way.
> >
> > But at the same point of time,
> > > Note that we do not claim VRS does not generalize beyond Bernoulli
> >
> > I am afraid Prop 1 and 2 are not enough to claim that it as a principaled method, I am not saying that it cannot be, but the empirical justification must be provided, and I encourage the authors to take this forward.
> >
> > Although, with the reclarifications, and writing changes, I will still update the score.

---

> ### Comment · Area_Chair_BMKj · 2025-08-07
> **Please respond to authors.**
>
> Dear Reviewer,
>
> Your active participation in the review process is crucial. Please respond to the authors' response and acknowledge you have done so.
>
> Thanks.
>
> -AC

---

> ### Author Response · Authors · 2025-08-07
>
> Thank you for the thoughtful follow-up and for taking the time to re-evaluate the submission, we truly appreciate it.
>
> We understand your concern about the generality of VRS beyond the Bernoulli case. We conducted a follow-up exploratory check during the initial rebuttal period to better understand how VRS performs in more complex discrete settings. Specifically, we applied VRS to Binomial distributions with varying $n \in $ { $1,2,3,4,5$ } and $p$, using a fixed Binomial proposal with $p=0.5$.  We observed that VRS consistently and noticeably reduced sampling bias compared to direct sampling, with trends aligning well with both the Bernoulli case and the behavior predicted by our extended versions of Propositions 1 and 2.
>
> To adhere to NeurIPS rebuttal guidelines as reminded by the AC, we are not including detailed results here, but we will incorporate them in the updated version of the paper. We believe these findings further support the potential generality of VRS as a prompting-based sampling framework, and we thank the reviewer for encouraging us to explore this direction.

---

### Author Response · Authors · 2025-08-09
**Summary of rebuttal discussion**

We thank the reviewers for their constructive feedback and for improving their scores (explicitly indicated by _Reviewer Q2Ka_ and _Reviewer VD8B_) following our clarifications. Below we summarize the main outcomes of the discussion and restate the contributions of our work.

---

### **Summary of Contributions**

* **Novel Method:** Verbalized Rejection Sampling (VRS), a natural-language adaptation of classical rejection sampling, which in our Bernoulli evaluation substantially reduces sampling bias compared to direct sampling across multiple LLMs.

* **Theory:** Bounds on total variation (TV) distance under model bias, separating gains from algorithmic structure versus prompt phrasing.

* **Conceptual Contribution:** Demonstrates how classical probabilistic algorithms can be embedded in natural language prompts for black-box LLMs, enabling structured probabilistic reasoning using natural language without tool-calling or model internals.


---

### **Summary of Discussion**

* **Methodological Clarification:** VRS is not few-shot prompting, but a stateless, algorithmic procedure expressed in natural language. Reviewers acknowledged this distinction after clarification.

* **Scope and Generalization:** While our current evaluation is in the Bernoulli setting, the VRS concept is general in principle, like classical rejection sampling. Extending it to other distributions is feasible but nontrivial, requiring distribution-specific empirical and theoretical analysis. In direct response to reviewer suggestions, we provided preliminary theoretical results and empirical findings for Binomial distributions, showing similar performance gains and applicability.

* **Tool-Free Motivation:** VRS is not proposed as a replacement for tool-calling, but to explore whether LLMs can simulate stochasticity internally in purely natural-language settings, a motivation recognized as valid and important by multiple reviewers.

* **Remaining Considerations:** Remaining points from the reviewers raised important forward-looking questions about how the VRS framework might scale to more complex distributions or settings.  These do not reflect concerns about the correctness, rigor, or contribution of the present work, but rather highlight **its potential to inspire follow-up research**. We believe this is a positive signal of the paper’s relevance and impact.

---

The discussion phase resolved early misunderstandings and confirmed the novelty, clarity, and rigor of our work. _The points still under discussion relate to future extensions and broader applicability, rather than limitations of the present work._ We hope this summary provides a clear view of the clarified scope and forward-looking significance of our contribution. We want to thank the reviewers and the Area Chair again for their time, engagement, and constructive feedback throughout this process.

---

### Decision · Program_Chairs · 2025-09-17

**Decision:**

Reject

**Comment:**

The primary contributions of this paper are (1) to analyze discrepancies between the Bernoulli distributions outputted by an LLM and a true Bernoulli distribution; and (2) a method that employs rejection sampling to get an LLLM to better approximate the true distribution. The reviewers note that the paper is well-written and easy-to follow, the combination of theoretical explanations and empirical evidence was nice to see, and the experiments were well designed to support the claims made in the paper.

While I agree with the reviewers that the paper is well-executed, I recommend "reject" due to the limited significance of the results. As reviewers brought up, the need for a computationally expensive method to get an LLM to approximate the Bernoulli distribution is not well-justified, especially in today's era, where state-of-the-art LLMs have been trained to run tools and write code.

Additional, the paper would also be more convincing if it included results arguments for distribution other than just Bernoulli. The authors' responses to reviewers include some preliminary analysis on approximating the binomial distribution, which I think will be a great addition to the paper. However, incorporating this analysis on binomial distributions will entail significant revisions to the text, which necessitate another round of reviews.

Finally, one reviewer pointed out the computational cost of the proposed method. The authors argued that is is unfair to do a cost comparison between traditional algorithms and LLMs because for traditional algorithms the wall-clock time (so-to speak) only starts after the solution has already been formalized into an algorithm, while for LLM inference, the formalization process is included in the wall-clock time (as the LLM reasons about its answer about its solution in natural language). I don't buy this argument. The process of a human formalizing a problem is not a computational cost since it is a human doing it, not the computer. Moreover, this process only happens once per type of problem, while the LLM needs to re-formalize the problem every time it's given a new instruction. I agree with the reviewer that the paper needs a deeper discussion of the computational cost of the proposed method compared to other methods.